# Microstructure and Corrosion Behavior of Zinc/Hydroxyapatite Multi-Layer Coating Prepared by Combining Cold Spraying and High-Velocity Suspension Flame Spraying

**DOI:** 10.3390/ma16206782

**Published:** 2023-10-20

**Authors:** Hailong Yao, Xiaozhen Hu, Qingyu Chen, Hongtao Wang, Xiaobo Bai

**Affiliations:** 1Jiangxi Province Engineering Research Center of Materials Surface Enhancing & Remanufacturing, School of Materials Science and Engineering, Jiujiang University, Jiujiang 332005, China; jjuchenqy@163.com (Q.C.); wanght7610@163.com (H.W.); 5784073@163.com (X.B.); 2School of Architecture Engineering and Planning, Jiujiang University, Jiujiang 332005, China; hxz5566@126.com

**Keywords:** Mg alloy, corrosion resistance, Zn/HA multi-layer, cold spraying, high-velocity suspension flame spraying

## Abstract

The study aims to enhance the corrosion resistance and bioactivity of Mg alloy substrates through the development of a zinc/hydroxyapatite multi-layer (Zn/HA-ML) coating. The Zn/HA-ML coating was prepared by depositing a cold-sprayed (CS) Zn underlayer and a high-velocity suspension flame sprayed (HVSFS) Zn/HA multi-layer and was compared with the CS Zn coating and the Zn/HA dual-layer (Zn/HA-DL) coating. Phase, microstructure, and bonding strength were examined, respectively, by X-ray diffraction, scanning electron microscopy, and tensile bonding testing. Corrosion behavior and bioactivity were investigated using potentiodynamic polarization, electrochemical impedance spectroscopy, and immersion testing. Results show that the HVSFS Zn/HA composite layers were mainly composed of Zn, HA, and ZnO and were well bonded to the substrate. The HVSFS HA upper layer on the CS Zn underlayer in the Zn/HA-DL coating exhibited microcracks due to their mismatched thermal expansion coefficient (CTE). The Zn/HA-ML coating exhibited good bonding within different layers and showed a higher bonding strength of 27.3 ± 2.3 MPa than the Zn/HA-DL coating of 20.4 ± 2.7 MPa. The CS Zn coating, Zn/HA-DL coating, and Zn/HA-ML coating decreased the corrosion current density of the Mg alloy substrate by around two–fourfold from 3.12 ± 0.75 mA/cm^2^ to 1.41 ± 0.82mA/cm^2^, 1.06 ± 0.31 mA/cm^2^, and 0.88 ± 0.27 mA/cm^2^, respectively. The Zn/HA-ML coating showed a sixfold decrease in the corrosion current density and more improvements in the corrosion resistance by twofold after an immersion time of 14 days, which was mainly attributed to newly formed apatite and corrosion by-products of Zn particles. The Zn/HA-ML coating effectively combined the advantages of the corrosion resistance of CS Zn underlayer and the bioactivity of HVSFS Zn/HA multi-layers, which proposed a low-temperature strategy for improving corrosion resistance and bioactivity for implant metals.

## 1. Introduction

Magnesium alloys exhibit some advantages of biodegradability, biocompatibility, and proper mechanical properties comparable to the human bone. However, rapid degradation rates limit their mechanical stabilities and their wider application in the field of implant materials [1]. In addition to alloying, purifying, and heat treatment, surface coatings hinder the penetration of a corrosion medium onto the surface of a Mg alloy substrate, which is one of the most widely used surface modifications to effectively improve the corrosion resistances of Mg alloys [2]. In addition to the corrosion resistance, the bioactivities of Mg alloys should also be improved to induce tissue differentiation and proliferation [3,4]. Therefore, a synergistic improvement of the corrosion resistance and the bioactivity is a key way to improve the biological applications of Mg alloys.

Functionally graded materials (FGMs) are innovative materials with compositions and/or microstructures gradually varying in space, which is a promising strategy for bio-metal substrates [5,6]. Graded coatings or multi-layer coatings can meet specific non-homogeneous service requirements without any abrupt interface at the macro-scale [7]. It can decrease mismatches of mechanical properties between the coatings and the substrates. There are two types of graded coatings or multi-layer coatings [7,8]:composition-graded coatings and microstructure-graded coatings. The composition-graded coatings present composition variations from the substrate to the coating surface, which is always composed of more than two different materials. Hydroxyapatite (HA) has been a well-known and well-studied biomaterial over the past few decades [9,10]. However, HA presents a significant mismatch of mechanical properties with the Mg alloy substrates, resulting in the peeling of the HA coating. HA/metal composites can relieve the mismatch between the ceramic coatings and the metal substrates, such as HA/Ta [11], HA/Ti [12], HA/Mg [13], and Ag/HA [14]. It should be noted that anti-corrosion coatings on Mg alloy substrates should also be degradable. Compared to many metals, Zn, Mg, and Fe are promising degradable biomaterials, and Zn has a proper corrosion rate between Fe and Mg [15]. Zn and its alloys have been considered promising candidates for surface coating Mg alloys. Therefore, the Zn/HA composite is considered to be a potential graded coating material on the Mg alloy, which can combine the advantages of both the HA and the Zn.

Functionally graded coatings or multi-layer coatings are successfully prepared by different technologies [7,16,17,18], i.e., thermal spraying, firing/enameling, electrophoretic deposition, and magnetron sputtering. It is worth noting that HA can be decomposed into calcium phosphate and calcium oxide at high temperatures [19], and HA decompositions are detrimental to its bioactivity and stability in the body [20]. Meanwhile, Zn has low melting and boiling points [15]. Commonly, each coating preparation technology presents advantages and disadvantages for different coating materials. Therefore, more than one method should be properly combined to prepare the Zn/HA-ML coating on a Mg alloy substrate.

Cold spray is an excellent technology for preparing metal coatings, which depends on the high-velocity impact of metal particles on the metal surface [21]. Cold-sprayed pure Zn and Zn alloy coatings effectively improve the anti-corrosion of the mild steel [22] and the Mg alloy [23]. Cold-sprayed Zn-HA/Zn double-layer coatings are successfully prepared and present excellent corrosion resistance on Mg alloys [23]. However, the pure Zn underlayer is thinned by the erosion of solid HA particles. Furthermore, compared with a pure Zn underlayer, the Zn/HA upper layer presents a negligible contribution to decreasing the corrosion current density. A similar phenomenon is also observed in the cold-sprayed HA/Ti composite coating [24] and the Al/A_2_O_3_ composite coating [25]. The 20 wt.% HA/Ti composite coating presents a higher corrosion current density than the pure Ti coating but exhibits a lower corrosion current density after annealing. A possible reason is the weak inter-particle bonding or microcracks in the HA/Ti composite coating [24], Al/A_2_O_3_ composite coatings [25], and Zn/HA composite coatings [26]. Therefore, it is necessary to explore a proper method of preparing Zn/HA composite layers on the cold-sprayed Zn underlayer to further improve corrosion resistance.

HA-based coatings are usually prepared by different technologies, such as plasma spraying [27], sol-gel processing [28], electrophoretic deposition [29], high-velocity oxy-fuel (HVOF) spray [30], aerosol deposition [31], and cold spraying [32]. During the preparation of Zn/HA composite coatings, the thermal decomposition of HA should be inevitable. Among these methods, high-velocity suspension flame spraying (HVSFS) is the proper method of preserving the HA phase stability [33]. It is reported that the HVSFS HA/Ti [12] and HA/Mg [13] composite coatings presented particle melting and proper bonding strengths. Although Ti particles and Mg particles are partly oxidized during HVSFS [12,13], it decreases mismatches between metal particles and HA particles. It is also reported that HVSF-sprayed HA/Mg composite coatings improve the corrosion resistance of the Mg alloy substrate in simulated body fluid [13]. However, there is no report on combining with cold spraying and high-velocity suspension flame spraying to prepare a Zn/HA-ML coating.

In the present study, a Zn/HA-ML coating was prepared on the AZ31B Mg alloy substrate, which was composed of a CS Zn underlayer and a HVSFS Zn/HA multi-layer. Phase, microstructure, bonding strength, corrosion behavior, and bioactivity of the Zn/HA-ML coating were investigated by comparing with the CS Zn coating and the Zn/HA dual-layer coating.

## 2. Experimental Procedures

### 2.1. Preparation of Cold-Sprayed Zn Coating

Zn powder (size 15–53μm, 98.6 wt.% Zn, 1.4 wt.% O, Beijing Youxinglian Nonferrous Metals Co., Ltd., Beijing, China) was used as the original material for Zn coating. A mixed powder of 50 vol.% Zn powder and 50 vol.% 1Cr18 shots (size range from 250 to 400 μm, ) was used for cold spraying, as shown in Figure 1a–c. Cast AZ31B Mg alloy plates (dimensions 10 mm ×10 mm ×4 mm and φ25.4 mm × 50 mm, Beijing Youxinglian Nonferrous Metals Co., Ltd., Beijing, China) were utilized as substrates. After being sand-blasted and cleaned in ethanol, Zn coatings, and Zn underlayer were deposited on AZ31B substrates by a cold-spraying system assembled by Xi’an Jiaotong University. The N_2_ primary gas pressure and temperature were 2.0 ± 0.1 MPa and 260 ± 20 °C, respectively. The spray distance was fixed to 20 mm, and the transverse speed of the torch was 50 mm/s.

### 2.2. Preparation of Zn/HA-DL Coating and Zn/HA-ML Coating

Nano-size HA powder (average size 20 nm, Beijing Deke Daojin Science And Technology Co., Ltd., Beijing, China) and Zn powder were the original materials for Zn/HA dual-layer coating and Zn/HA multi-layer coating. Three types of mixed particle suspensions (HA, Z7H3, and Z3H7) were prepared by mixing HA powders with Zn powders at the weight proportions of 10, 3:7, and 7:3, respectively. Different mixed particle suspensions were prepared by dispersing 10 wt.% of pure HA powders or Zn/HA mixed powders into 90 wt.% of solvent. The solvent was composed of distilled water and ethanol in a 1:1 weight proportion. Different particle suspensions were fully mixed for at least 10 h by a mechanical stirrer.

An HVSFS system was applied to deposit the pure HA layer and the Zn/HA composite layers, as shown in Figure 1d–e. For the Zn/HA-DL coating, a pure HA upper layer was deposited on the CS Zn underlayer by the HVSFS system. For the Zn/HA-ML coating, a Z7H3 layer, a Z3H7 layer, and an HA upper layer were sequentially deposited on the CS Zn underlayer by the HVSFS system. The HVSFS system was a modified gas-fuel HVOF torch (CH2000, Xi’an Jiaotong University, Xi’an, China), as shown in Figure 1d. The procedure was performed using an axial cylindrical suspension injection nozzle with a 0.5 mm-diameter orifice instead of a standard dry powder injection. The particle suspensions were mechanically fed into the combustion chamber during coating depositions. The spraying parameters of HVSFS are listed in Table 1.

### 2.3. Characterization of Phase and Microstructure

Phase structures of different coatings were identified by X-ray diffraction (XRD, D8 Advance, Bruker, Salbrücken, Germany) at Cu_Kα_ radiation of 1.5418 Å, operating voltage of 35 kV, operating current of 35 mA, 2θ range of 20–90°, and a scan rate of 0.1 °/s. Surface morphology and cross-sections of CS Zn coating, Zn/HA-DL coating, and multi-layer coating, as well as after immersion, were investigated by scanning electron microscopy (SEM, VEGA II, Tescan, Brno, Czech Republic) with energy-dispersive spectrometry (EDS). Phase compositions of the Zn/HA-ML coating before and after immersion were examined by Fourier-transform infrared spectroscopy (FTIR, MX-1E, Nicolet, Thermo Fish, Waltham, MA, American) using KBr pellets technology with a scan range of 400–4000 cm^−1^.

### 2.4. Bonding Strength Test

To investigate the bonding strength of the CS Zn coating, the Zn/HA-DL coating, and the Zn/HA-ML coating, a standard tensile test method (ASTM C633) [34] was applied. Different coatings were bonded on grit-blasted facings of the loading fixtures, being the same size and shape as the sample. An adhesive glue with a tensile strength of about 70 MPa (E-7, Adtest, Shanghai huayi resins Co., Ltd., Shanghai, China) was used. The assembly was held perpendicularly and placed in a muffle furnace at 100 °C for 2 h. The assembly was loaded in a tensile testing machine at a head speed of 1 mm/min. Bonding strengths for different coatings were the average value of at least three tests. Fractured surfaces after tensile testing were investigated by scanning electron microscopy.

### 2.5. Corrosion and Immersion Testing

To evaluate the corrosion behaviors of different coatings, potentiodynamic polarization testing and electrochemical impedance spectroscopy (EIS) were applied. Before electrochemical testing, different coatings were immersed in simulated body fluid (SBF) at 37 ± 1 °C for 1, 7, and 14 days. During immersion testing, different coatings on the substrate with an area of 1 cm^2^ were exposed to simulated body fluid (Modified-SBF, Jisskang, Qingdao, China), and other surfaces were covered by epoxy resin. The SBF contained 142.0 mM Na^+^, 5.0 mM K^+^, 1.5 mM Mg^2+^, 2.5 mM Ca^2+^, 103.0 mM Cl^−^, 10.0 m HCO_3_^−^, 1.0 mM HPO_4_^2−^, and 0.5 mM SO_4_^2−^. During immersion, the volume of SBF followed the equation V_s_ = S_a_/10 [35], where v_s_. is the volume of SBF (mL) and S_a_ is the apparent surface area of the specimen (mm^2^). During electrochemical testing, a flat three-electrode cell consisting of a working electrode (coating or bare substrate), a reference electrode (Ag/AgCl electrode), and a counter electrode (Pt plate) was utilized. Polarization curves were carried out on an electrochemical workstation (IM6, Zahner, Kronach, Germany) at open circuit potential from −0.5 V to 0.5 V (vs. SCE) with a scan rate of 1 mV/s. CorrWare software (CView 2, version 2.6) was employed to estimate the corrosion potential (Ecorr), corrosion current density (Icorr), and corrosion rate. EIS measurements were carried out at open-circuit potential (OCP) with an amplitude of 10 mV and with a scanning frequency from 0.01 Hz to 100 kHz. Recorded impedance spectra were fitted by using Z-view software (ZView 2, version 3.0a).

## 3. Results and Discussion

### 3.1. Phase Structure

Figure 2 shows XRD patterns of the CS Zn underlayer and the HA layer, the Z7H3 composite layer, and the Z3H7 composite layer deposited by HVSFS. It can be seen that the CS Zn underlayer was composed of Zn and ZnO, and ZnO was formed due to its low melting point and high impact energy [36]. The HVSFS HA layer exhibits no decompositions. Two HVSFS Zn/HA composite layers were composed of Zn, HA, and ZnO. Some of the Zn particles were oxidized to ZnO in the presence of atmospheric air [37], and the peak intensities of ZnO and Zn were increased with the Zn content in mixed powders for the Z7H3 composite layer and the Z3H7 composite layer. Although the Zn particles were partly oxidized to ZnO during cold spraying and HVSFS, ZnO preserves many qualities, such as biocompatibility and antibacterial [38]. It is reported that TiO_2_ could enhance the HA decomposition [12], but the ZnO could not react with the HA in the Zn/HA composites [37]. The phase stability of the HVSFS HA layer was also ascribed to the low-temperature flame during HVSFS and the presence of a water stream associated with the presence of distilled water in the suspension and the suitable oxy-fuel ratio [12]. This result indicates that proper Zn/HA composite layers could be formed by HVSFS.

### 3.2. Cross-Sectional Microstructures

Figure 3 shows cross-sections of the CS Zn coating, Zn/HA-DL coating, and Zn/HA-ML coating. All the coatings were well bonded to the Mg alloy substrate, especially in the Zn/HA-DL coating and the Zn/HA-ML coating. In Figure 3a, the Zn coating presented a good bond to the Mg alloy substrate, and no obvious microcracks and holes were observed in the CS Zn underlayer with irregular and flattened Zn particles [23]. In the Zn/HA-DL coating, although the HVSFS HA upper layer was bonded to the Zn underlayer, some microcracks existed in the HA upper layer (red arrows in Figure 3b). It is reported that the thermal expansion coefficient (CTE) of the Mg alloy, Zn, and HA is 26 × 10^−6^/°C [39], 30 × 10^−6^/°C [40], and 10.6 × 10^−6^/°C [41], respectively. Microcracks in the HVSFS HA upper layer could be mainly attributed to the CTE mismatch between the Zn underlayer and the HA upper layer.

In the Zn/HA-ML coating, the CS Zn underlayer, the HVSFS Zn/HA composite layers, and the pure HA upper layer were obviously differentiated, as shown in Figure 3c. It is easy to discern that gray regions refer to the HA phase, and light regions refer to the Zn phase. Light regions and gray regions were alternately stacked in the Zn/HA composite layers. The Zn/HA composite layers could effectively relieve the CTE mismatch between the CS Zn underlayer and the HVSFS HA upper layer, and obvious microcracks were not observed in the Zn/HA-ML coating, as shown in Figure 3c. This phenomenon is observed in many graded coatings deposited by thermal spraying technology [42,43] and cold-spraying technology [44]. Although the Zn/HA composite layers were deposited by different original Zn/HA mixed powers, the content of Zn in the Zn/HA composite layer was higher than HA. The possible reason could be attributed to the difference in size, density, and melting point between Zn particles and HA particles. During co-depositions of Zn particles and HA particles, nano-sized HA powders were subjected to flow with the flame rather than impact onto the substrate [45], while large and melt Zn powders could be prior to deposit onto the substrate. A similar phenomenon is also observed in that HVSFS Ti/HA composite coatings were mainly composed of Ti rather than HA regardless of Ti/HA ratios [12]. Compared to the CS Zn/HA composite layer [23] and the Ta/HA composite layer [11], the HVSFS Zn/HA composite layer presented a proper thickness and particle accumulation on the CS Zn underlayer. Therefore, it can be considered that combining cold spraying and high-velocity suspension flame spraying could prepare a Zn/HA-ML coating on the Mg alloy substrate.

### 3.3. Bonding Strength

Tensile strengths of the CS Zn coating, Zn/HA-DL coating, and Zn/HA-ML coating were measured as shown in Figure 4. The bonding strength of the CS Zn coating, the Zn/HA-DL coating, and the Zn/HA-ML coating were 42.1 ± 1.5 MPa, 20.4 ± 2.7 MPa, and 27.3 ± 2.3 MPa, respectively. Both the Zn/HA-DL coating and the Zn/HA-ML coating showed lower bonding strengths than the plasma-sprayed Ti/HA dual-layer coating [46] and the cold-sprayed Ta/HA dual-layer coating [11] but were comparable to the plasma-sprayed HA coating [47] and higher than the cold-sprayed HA coating [26]. Meanwhile, this bonding strength was high enough to meet the requirement of the international standard ISO 13779-2 [48], which prescribes the bond strength in excess of 15 MPa for APS HA coatings. It is worth noting that the Zn/HA-ML coating exhibited a higher bonding strength than that of the Zn/HA-DL coating but lower than the CS Zn coating. This could be attributed to the significant difference in mechanical properties between different layers.

Fractured surfaces of different coatings were examined, as shown in Figure 4i–iii. The fractured surface of the CS Zn coating exhibited smooth regions (blue arrows) and tear ridges (yellow arrows) within the Zn coating, as shown in Figure 4i, which are the typical fracture features of cold-sprayed metal deposits [49]. It indicates that the adhesion strength is higher than the cohesion strength in the cold-sprayed Zn coating [46]. Figure 4ii exhibits the HA phase (light regions) and the Zn phase (gray regions) on the fractured surface for the Zn/HA-DL coating. Furthermore, some spherical Zn particles were also observed on the fractured surface (hollow red arrows in Figure 4ii). Although there were no large pores in the HVSFS HA upper layer for the Zn/HA-DL coating, vertical and parallel micro-cracks were obvious, as shown in Figure 3b. This result indicates the fracture mainly occurred at the interface between the CS Zn underlayer and the HVSFS HA upper layer. Figure 4iii also exhibits both HA particles and Zn particles on the fractured surface for the Zn/HA-ML coating. It indicates that the fracture occurred within the Zn/HA composite layer due to the decrease in CTE mismatch [46]. This result further confirms that the HVSFS Zn/HA composite layers could be effectively bonded with both the CS Zn underlayer and the HVSFS HA upper layer.

### 3.4. Potentiodynamic Polarization Property

To examine the corrosion behaviors of different coatings, potentiodynamic polarization was measured after different immersion times. Figure 5 shows the polarization curves of the CS Zn coating, Zn/HA-DL coating, and Zn/HA-ML coatings after immersion in SBF for 1 day, 7 days, and 14 days, respectively. The Tafel extrapolation method [50] was applied to fit the polarization curves and corrosion potential (*E*_corr_), corrosion current density (*I*_corr_), and corrosion rate. The Tafel slopes are listed in Table 2. The corrosion rate is calculated according to ASTM G102-2015 [51] by following the equation CR = (*I*_corr_ × K × EW)/(d × A), in which *I*_corr_ is corrosion current, K is constant, EW is equivalent weight, d is density, and A is area. It can be seen that the CS Zn coating, Zn/HA-DL coating, and Zn/HA-ML coating exhibited higher *E*_corr_, lower *I*_corr_, and lower corrosion rates than the bare Mg alloy substrate after immersion of 1 day. Compared to the bare substrate, the decrease in both *I*_corr_ and corrosion rate of different coatings was around threefold and twofold, respectively. It indicates the prior corrosion resistance of different coatings could be attributed to the high anti-corrosion of Zn coating [22]. Compared with the CS Zn coating, the Zn/HA-DL coating and Zn/HA-ML coating further decreased the *I*_corr_ from 1.41 ± 0.82mA/cm^2^ to 1.06 ± 0.31 mA/cm^2^ and 0.88 ± 0.27 mA/cm^2^ and decreased the corrosion rate from 16.5 ± 3.6 mm/year to 12.5 ± 3.7 mm/year and 10.3 ± 3.2 mm/year, respectively. This result confirms that the HVSFS HA upper layer and the HVSFS Zn/HA composite layers improved corrosion resistances of the CS Zn underlayer, which could be mainly related to the lower conductivity of HA.

By prolonging the immersion time to 7 days and 14 days, the CS Zn coating, Zn/HA-DL coating, and Zn/HA-ML coating exhibited lower *I*_corr_ and corrosion rates, although the corrosion potentials were stable. Compared with an immersion time of 1 day, the CS Zn coating after an immersion time of 14 days exhibited a decrease in *I*_corr_ and corrosion rate by around twofold from 1.41 ± 0.82 mA/cm^2^ to 0.73 ± 0.12 mA/cm^2^ and from 16.5 ± 3.6 mm/year to 8.5 ± 1.2 mm/year, respectively. The Zn/HA-DL coating after an immersion time of 14 days exhibited a decrease in *I*_corr_ and corrosion rate by around threefold from 1.06 ± 0.31 mA/cm^2^ to 0.37 ± 0.12 mA/cm^2^ and from 12.5 ± 3.7 mm/year to 4.4 ± 1.5 mm/year, respectively. The Zn/HA-ML coating after an immersion time of 14 days exhibited a decrease in *I*_corr_ and corrosion rate by around sixfold from 0.88 ± 0.27 mA/cm^2^ to 0.14 ± 0.07 mA/cm^2^ and from 10.3 ± 3.2 mm/year to 1.7 ± 0.9 mm/year, respectively. The improvement in anti-corrosion properties after immersion could be mainly attributed to the formation of corrosion byproducts [52]. This result indicates that the HVSFS HA upper layer and Zn/HA middlelayer further improved the corrosion resistance of the CS Zn coating after immersion, and the HVSFS Zn/HA middlelayer was prior to the HVSFS HA layer.

### 3.5. Electrochemical Impedance Spectroscopy (EIS)

To further investigate the corrosion behaviors of different coatings, the EIS was measured and is shown in Figure 6. Due to the good bonding, the interface between the Zn coating and Mg alloy substrate can be considered as an ohmic contact. Semicircles in impedance spectroscopy can refer to the charge transfer resistance of the substrate/electrolyte coating/electrolyte interface [24]. An equivalent circuit diagram was applied to fit Nyquist impedance plots, as shown in Figure 6a. For the bare substrate, *R*_s_ represents the ohmic resistance of the substrate and conductive lines, and *R*_1_ and *CPE*_1_ represent the charge transfer resistance and capacitive characteristics at the bare substrate/electrolyte interface. For different coatings, according to the literature [13,22,23], *R*_1_ and *CPE*_1_ represent the charge transfer resistance and capacitive characteristics at the dense coating/electrolyte interface, respectively; *R*_2_ and *CPE*_2_ represent the charge transfer resistance and capacitive of the porous Zn/HA composite layer/electrolyte interface or HA/electrolyte interface, respectively. Fitting results of EIS for different coatings are listed in Table 3.

It can be found that all the coatings exhibited higher corrosion resistance than the bare substrate and improvements in corrosion resistance after immersion, which was consistent with the results of polarization properties. Compared to the bare Mg alloy substrate, all the coatings exhibited higher polarization resistances (*R*_p_ = *R*_1_ + *R*_2_) [53]. Polarization resistance (*R*_p_) of the CS Zn coating, the Zn/HA-DL coating, and the Zn/HA-ML coating after an immersion time of 1 day were higher than that of the bare substrate by around sixfold, sixteenfold, and elevenfold, respectively. With the extension of immersion time to 14 days, the *R*_p_ of all the coatings was further increased by around one–threefold. It can be seen that the *R*_p_ of the Zn/HA-DL coating and the Zn/HA-ML coating was higher than the CS Zn coating after different immersion periods. This result indicates that the HVSFS HA upper and the Zn/HA composite layers contributed to the improvement in corrosion resistance of the CS Zn coating.

It is reported that the cold-sprayed Zn/HA [23] and Ti/HA [24] composite layer contributed to a limited improvement in the corrosion resistance of the cold-sprayed pure metal coating. The possible reason is the weak bonding and microcracks between the metal particles and ceramic particles induced by solid formation during cold spraying. The corrosion medium could penetrate into the inner coating or the substrate through weak interfaces and microcracks [24]. During HVSFS, metal or ceramic particles are fully and partially melted and oxidized [12], which could enhance the interface bonding between metal and ceramic particles. With the extension of the immersion time, tiny pores or microcracks within coatings could be sealed by corrosion by-products of metal particles or newly formed depositions, resulting in an increase in corrosion resistance [52,54]. Therefore, all the coatings showed improvements in corrosion resistance after immersion, especially for the Zn/HA-DL coating and the Zn/HA-ML coating.

### 3.6. Microstructures after Immersion

Figure 7 shows surface microstructures of the CS Zn coating and the Zn/HA-ML coating before and after immersion for 14 days. The CS Zn coating before immersion exhibited a rough surface and large craters (white dotted line) with irregular, flattened, and spherical Zn particles (hollow red arrows), as shown in Figure 7a. The craters were induced by large stainless steel shots during cold spraying [55]. The CS Zn coating exhibited some melt features (blue arrows in Figure 7a) due to impact-induced melting during cold spraying [36]. After depositing the Zn/HA-ML coating by HVSFS, the craters disappeared, and the typical surface was composed of smooth, flattened, spherical, and nano-agglomerating HA particles (black and white arrows in Figure 7c). This surface feature of nano-sized HA particle deposition by HVSFS was consistent with our previous results [12].

After immersion for 14 days, the CS Zn coating was covered by some clusters with tiny particles (yellow arrows in Figure 7b). EDS results show that the clusters were mainly composed of Zn, Ca, P, Cl, and O, which contributed to the formation of corrosion byproducts of Zn coating [56] and a small residue of corrosion mediums. In Figure 7d, tiny spherical particles (red arrows) and clusters with tiny particles (green arrows) were observed without any smooth flattened features on the Zn/HA-ML coating after immersion for 14 days. EDS results show that the Zn/HA-ML coating after immersion was mainly composed of Zn, Ca, P, Cl, and O. Compared to Zn/HA-ML coating before immersion, it indicates that Zn particles in Zn/HA middle-layers were also dissolved, and a small number of Zn ions diffused onto the surface of the Zn/HA-ML coating.

Figure 8 shows a typical cross-section of the Zn/HA-ML coating after immersion for 14 days. There was no trace of corrosion mediums penetrating the substrate through the coating (Figure 8i), which further confirms the corrosion protection of the Zn/HA-ML coating. In Figure 8ii, a layer of newly formed apatite and corrosion by-products could be observed, tabbed as a yellow dotted line. Some spherical particles (yellow arrows) within the newly formed layer were consistent with the surface features in Figure 7d. In the Zn/HA composite layer (Figure 8iii), in addition to the Zn phase (blue arrows) and the HA phase (white arrows), the corrosion by-products were observed around the Zn particles (red arrows). According to electrochemical results from Figure 5 and Figure 6, it can be considered that the corrosion by-products around Zn particles and newly formed apatite depositions contributed to improvements in the corrosion resistance of the Zn/HA-ML coating after immersion [52]. Meanwhile, the FTIR spectrum of the Zn/HA-ML coating before and after immersion revealed that PO_4_^3−^ and OH^−^ were the main functional groups, as shown in Figure 9. The formation of new apatite was common on thermal-sprayed HA-based coatings after immersion in SBF [33]. Combined with electrochemical results, the Zn/HA-ML coating not only increased the corrosion resistance of the Mg alloy substrate but also improved its bioactivity. Compared with plasma spraying [27] and HVOF [30], the Zn/HA-ML coating was prepared by low-temperature technologies of cold spray and high-velocity suspension flame spray technology. Furthermore, original nano-sized HA particles could be directly used as feedstock. This Zn/HA-ML coating combined exhibited a potential for other temperature-sensitive and nano-sized materials.

## 4. Conclusions

In this study, a Zn/HA-ML coating was prepared on a Mg alloy substrate by combining cold spraying and high-velocity suspension flame spraying. HVSFS Zn/HA composite layers were mainly composed of Zn and HA in addition to ZnO, and the HVSFS HA upper layer was not decomposed. The HVSFS HA upper layer in the Zn/HA-DL coating presented microcracks due to the CTE mismatch. The Zn/HA-ML coating effectively decreased the CTE mismatch and showed a higher bonding strength of 27.3 ± 2.3 MPa than the Zn/HA-DL coating of 20.4 ± 2.7 MPa by 30%. Both the CS Zn coating, the Zn/HA-DL coating, and the Zn/HA-ML coating decreased corrosion current density of the Mg alloy substrate by around two–fourfold from 3.12 ± 0.75 mA/cm^2^ to 1.41 ± 0.82mA/cm^2^, 1.06 ± 0.31 mA/cm^2^, and 0.88 ± 0.27 mA/cm^2^, respectively, and exhibited significant decreases in corrosion current density to 0.73 ± 0.12 mA/cm^2^, 0.37 ± 0.12 mA/cm^2^, and 0.14 ± 0.07 mA/cm^2^ after an immersion time of 14 days. The Zn/HA-ML coating after an immersion time of 14 days showed more decreases in the corrosion current density by sixfold and more improvements in the corrosion resistance by twofold, which was mainly attributed to newly formed apatite and corrosion by-products of Zn coatings on/within the coating. This Zn/HA-ML coating effectively combined the advantages of the corrosion resistance of the CS Zn layer and bioactivities of the HVSFS Zn/HA composite layers, which proposed a low-temperature strategy for improving corrosion resistance and bioactivities.

## Figures and Tables

**Figure 1 materials-16-06782-f001:**
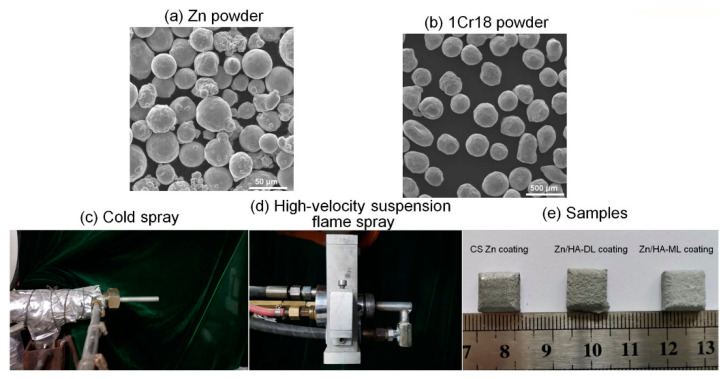
Morphologies of powders, experimental facilities, and samples.

**Figure 2 materials-16-06782-f002:**
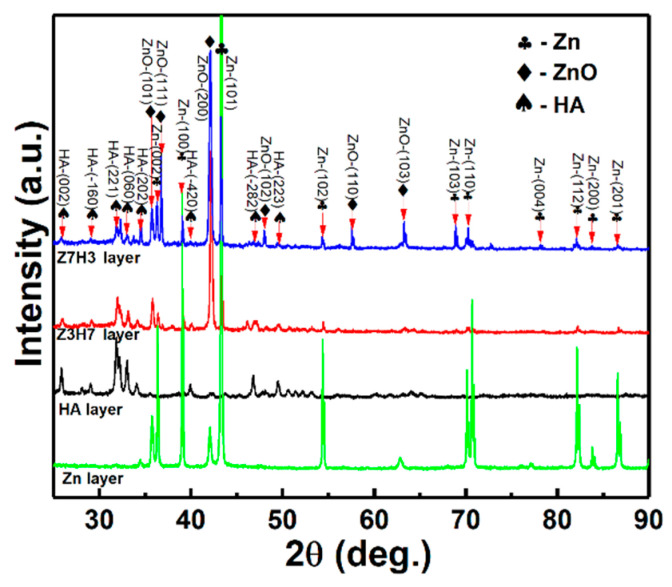
XRD patterns of CS Zn coating, Zn/HA-DL coating, and Zn/HA-ML coating.

**Figure 3 materials-16-06782-f003:**
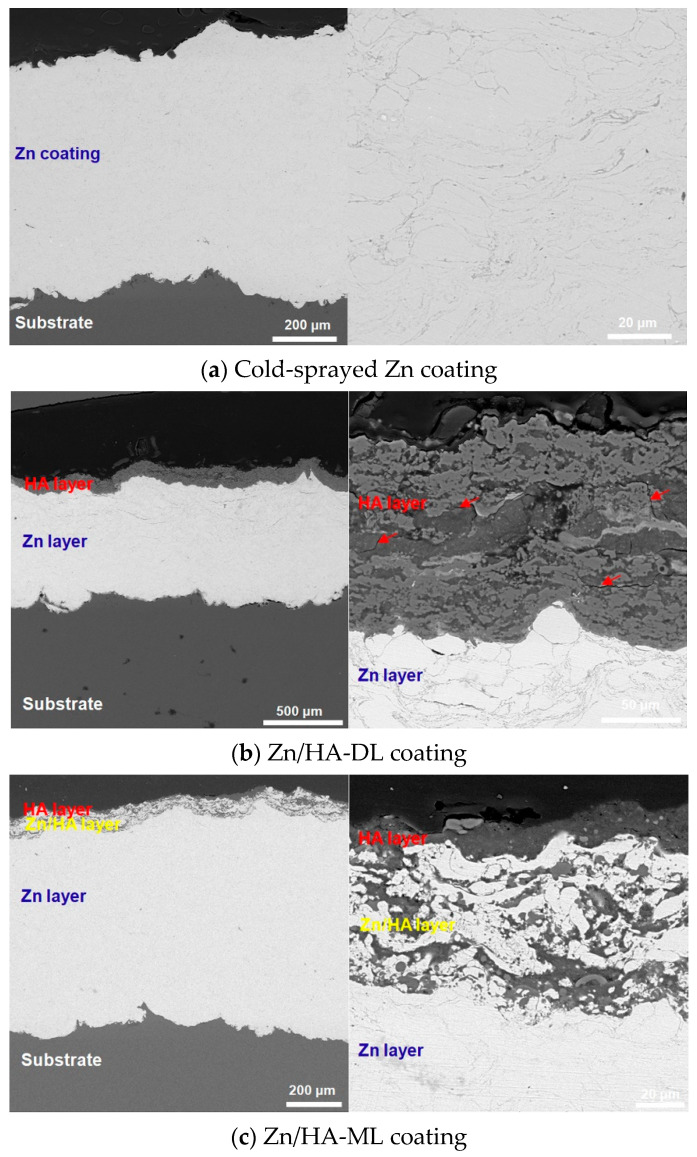
Cross-sections of CS Zn coating, Zn/HA-DL coating, and Zn/HA-ML coatings. (**a**) CS Zn coating, (**b**) Zn/HA-DL coating, (**c**) Zn/HA-ML coating.

**Figure 4 materials-16-06782-f004:**
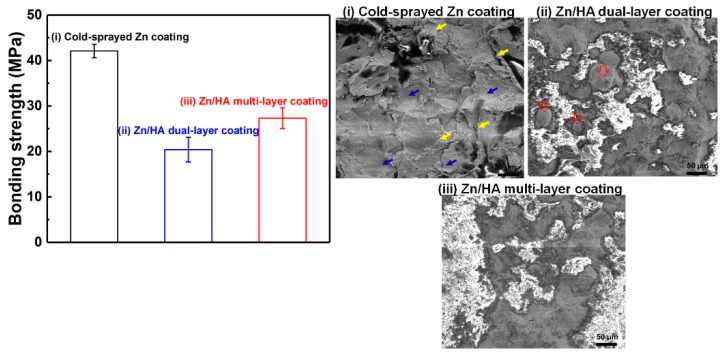
Bonding strength and fractured surfaces of CS Zn coating, Zn/HA-DL coating, and Zn/HA-ML coating (error bars were the standard deviation of more than three tests for each condition).

**Figure 5 materials-16-06782-f005:**
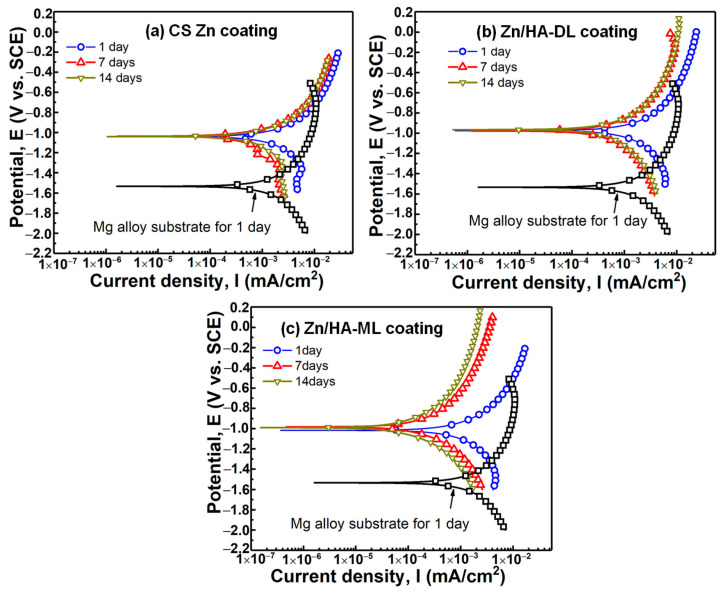
Polarization curves of CS Zn coating, Zn/HA-DL coating, and Zn/HA-ML coating. (**a**) CS Zn coating, (**b**) Zn/HA-DL coating, (**c**) Zn/HA-ML coating.

**Figure 6 materials-16-06782-f006:**
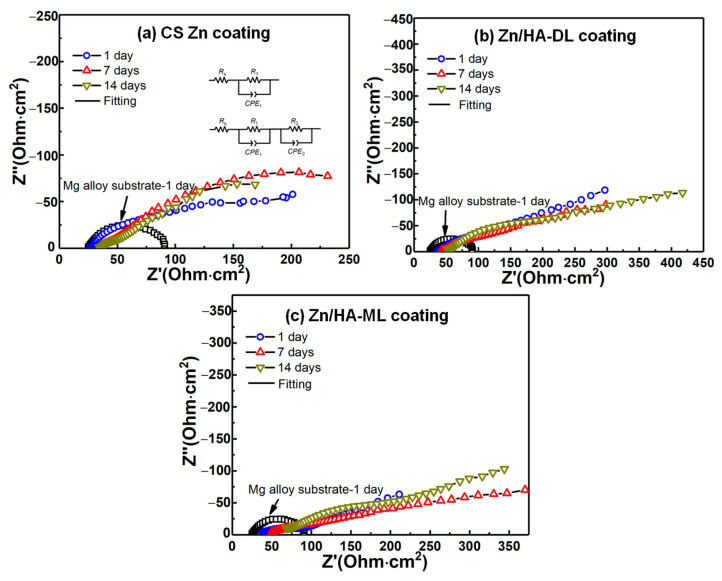
EIS plots of CS Zn coating, Zn/HA-DL coating, and Zn/HA-ML coating. (**a**) CS Zn coating, (**b**) Zn/HA-DL coating, (**c**) Zn/HA-ML coating.

**Figure 7 materials-16-06782-f007:**
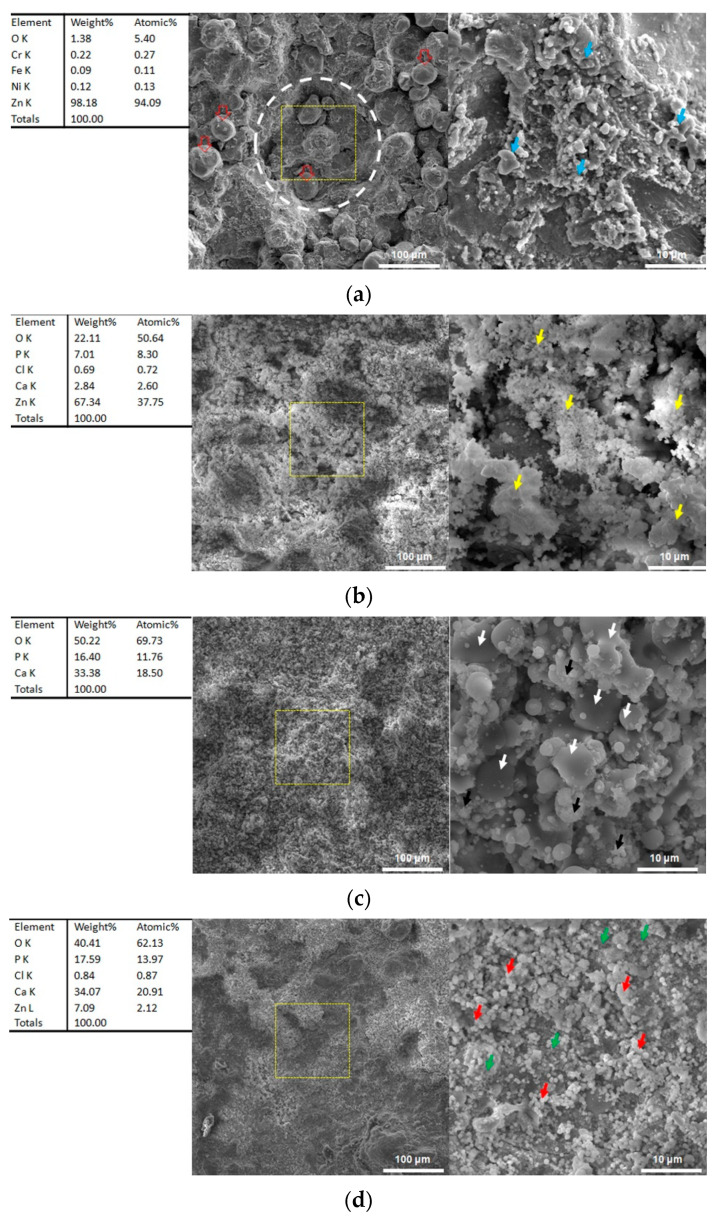
Surface morphologies of CS Zn coating and Zn/HA-ML coating before and after immersion of 14 days. (**a**) CS Zn coating before immersion, (**b**) CS Zn coating after immersion, (**c**) Zn/HA-ML coating before immersion, (**d**) Zn/HA-ML coating after immersion.

**Figure 8 materials-16-06782-f008:**
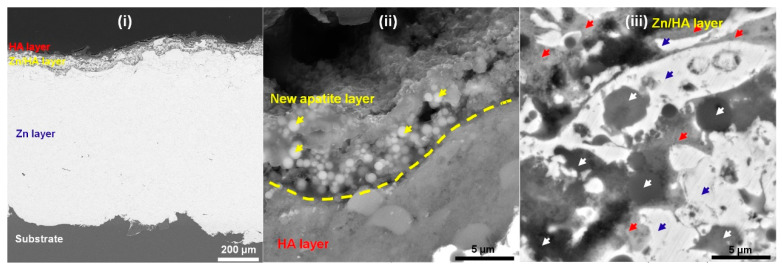
Typical cross-section of Zn/HA-ML coating after an immersion time of 14 days. (**i**) Cross-section, (**ii**) New apatite layer, (**iii**) Zn/HA composite layer.

**Figure 9 materials-16-06782-f009:**
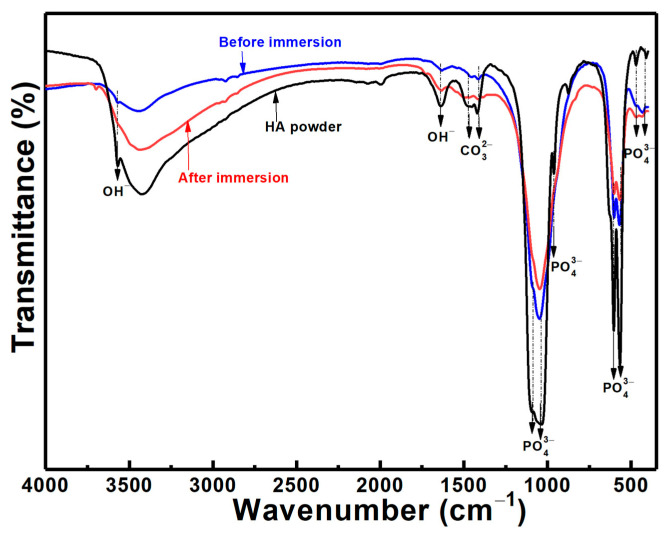
FTIR of Zn/HA-ML coatings before and after an immersion time of 14 days.

**Table 1 materials-16-06782-t001:** Spraying parameters of HVSFS.

Fuel (propane) flow rate, slpm	27
Oxygen flow rate, slpm	181
Suspension flow rate, mL/min	40
Spray distance, mm	100
Torch traverse speed, mm/s	500

**Table 2 materials-16-06782-t002:** Fitting results of polarization curves in Figure 5 (standard deviations originated from more than three tests for each condition).

Samples	Immersion Time (Day)	*β*_α_ (mV)	*β*_c_ (mV)	*I*_corr_ (mA/cm^2^)	*E*_corr_ (V)	Corrosion Rate (mm/Year)
AZ31B sub.	1	690.6 ± 59.9	417.6 ± 144.3	3.12 ± 0.75	−1.54 ± 0.01	29.9 ± 0.9
CS Zn coating	1	455.8 ± 29.7	265.5 ± 82.3	1.41 ± 0.82	−1.03 ± 0.02	16.5 ± 3.6
	7	678.4 ± 43.3	412.9 ± 281.8	0.95 ± 0.52	−1.04 ± 0.06	11.2 ± 4.2
	14	517.5 ± 45.1	314.8 ± 109.6	0.73 ± 0.12	−1.11 ± 0.09	8.5 ± 1.2
Zn/HA-DL coating	1	413.2 ± 60.1	291.1 ± 36.8	1.06 ± 0.31	−0.94 ± 0.01	12.5 ± 3.7
	7	451.1 ± 32.7	312.1 ± 20.8	0.56 ± 0.04	−0.97 ± 0.01	6.7 ± 0.6
	14	368.3 ± 47.3	283.7 ± 17.9	0.37 ± 0.12	−0.97 ± 0.02	4.4 ± 1.5
Zn/HA-ML coating	1	417.2 ± 17.3	336.1 ± 25.1	0.88 ± 0.27	−0.99 ± 0.02	10.3 ± 3.2
	7	296.7 ± 19.5	311.5 ± 45.8	0.24 ± 0.11	−0.98 ± 0.01	2.8 ± 1.4
	14	281.81 ± 67.3	318.5 ± 16.1	0.14 ± 0.07	−0.99 ± 0.03	1.7 ± 0.9

**Table 3 materials-16-06782-t003:** Fitting results of EIS in Figure 6 (standard deviations originated from more than three tests for each condition).

Samples	Immersion Time (Day)	*R_s_* (ohm·cm^2^)	*R*_1_ (ohm·cm^2^)	*CPE*_1_ (mF·cm^−2^·sn)	n_1_	*R*_2_ (ohm·cm^2^)	*CPE*_2_ (mF·cm^−2^·sn)	n_2_	*R_p_* (*R*_1_ + *R*_2_) (ohm·cm^2^)
AZ31B sub.	1	27.2 ± 5.6	-	-	-	21.5 ± 5.8	0.0511 ± 0.0112	0.767 ± 0.019	21.5 ± 5.8
CS Zn coating	1	26.8 ± 1.1	36.5 ± 19.2	0.0017 ± 0.0008	0.832 ± 0.071	100.2 ± 42.6	0.0121 ± 0.0014	0.607 ± 0.131	136.7 ± 51.2
	7	30.9 ± 4.28	25.9 ± 12.4	0.0021 ± 0.0006	0.722 ± 0.149	163.6 ± 50.4	0.0151 ± 0.0062	0.486 ± 0.117	189.5 ± 78.1
	14	33.9 ± 8.9	57.1 ± 7.1	0.0058 ± 0.0007	0.503 ± 0.142	323.3 ± 91.6	0.0121 ± 0.0045	0.546 ± 0.288	380.4 ± 88.6
Zn/HA-DL coating	1	46.7 ± 10.4	118.2 ± 69.5	0.0013 ± 0.0005	0.616 ± 0.058	241.4 ± 89.1	0.0182 ± 0.0015	0.417 ± 0.359	359.5 ± 85.6
	7	52.1 ± 4.1	102.2 ± 49.4	0.0011 ± 0.0002	0.582 ± 0.056	358.6 ± 96.4	0.0153 ± 0.0013	0.596 ± 0.081	460.8 ± 94.8
	14	60.4 ± 3.4	239.8 ± 86.6	0.0012 ± 0.0004	0.612 ± 0.111	365.2 ± 83.5	0.0145 ± 0.0082	0.731 ± 0.336	604.9 ± 81.8
Zn/HA-ML coating	1	30.8 ± 6.9	76.7 ± 23.8	0.0019 ± 0.0004	0.374 ± 0.012	160.7 ± 56.8	0.0246 ± 0.0091	0.668 ± 0.013	237.4 ± 65.3
	7	52.2 ± 10.1	107.8 ± 36.2	0.0011 ± 0.0007	0.543 ± 0.043	193.8 ± 63.9	0.0177 ± 0.0072	0.736 ± 0.143	301.6 ± 89.2
	14	66.1 ± 9.8	184.3 ± 62.1	0.0015 ± 0.0001	0.513 ± 0.032	292.9 ± 62.1	0.0147 ± 0.0011	0.714 ± 0.121	477.2 ± 78.9

## Data Availability

Data will be made available on request.

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
