# Peer review of "Microstructure and Corrosion Behavior of Zinc/Hydroxyapatite Multi-Layer Coating Prepared by Combining Cold Spraying and High-Velocity Suspension Flame Spraying"

_materials, 2023, doi:10.3390/ma16206782_

Round 1

Reviewer 1 Report

The manuscript addresses a problem of great importance in the health area regarding the development of prostheses using magnesium alloys. Protection of the magnesium alloy is required to prevent rapid corrosion.

The application of layers of Zn and hydroxyapatite is tested, showing that the best results are obtained with the use of multilayers. The behavior of the protective layers is evaluated by different methodologies, including polarization curves and electrochemical impedance.

It is recommended that in Figures 5 and 6 the blank results for magnesium alloy be included in the three images (a, b and c). When comparing tables 2 and 3, it is observed that there is a difference in the corrosion behavior between Zn/HA dual-layer coating and Zn/HA multi-layer coating.

It is recommended that a possible explanation be offered for the difference.

Author Response

Response to Reviewer#1

Thank you for your valuable comments. We revised the manuscript according to your specific comments as follows. All corrections were underlined, which you could easily find in the revised article. The following is responses and explanations to your comments.

Comments:

The manuscript addresses a problem of great importance in the health area regarding the development of prostheses using magnesium alloys. Protection of the magnesium alloy is required to prevent rapid corrosion.

The application of layers of Zn and hydroxyapatite is tested, showing that the best results are obtained with the use of multilayers. The behavior of the protective layers is evaluated by different methodologies, including polarization curves and electrochemical impedance.

It is recommended that in Figures 5 and 6 the blank results for magnesium alloy be included in the three images (a, b and c). When comparing tables 2 and 3, it is observed that there is a difference in the corrosion behavior between Zn/HA dual-layer coating and Zn/HA multi-layer coating. It is recommended that a possible explanation be offered for the difference.

Responses: Thank you for your kind indications. The blank results for magnesium alloy have been added in Figure 5 and Figure 6.

The difference in the corrosion behavior between Zn/HA-DL coating and Zn/HA-ML coating could be attributed to the microstructures and compositions of the upper layer. On the one hand, the Zn/HA-DL coating presented thicker pure HA layer than that the pure HA in the Zn/HA-ML coating. Compared to Zn, HA is insulation. On the other hand, pure HA layer in Zn/HA-DL coating exhibited some microcracks due to the TEC mismatch. Corrosion electrolyte could penetrate into the Zn under-layer. The Zn/HA-ML coating presented denser microstructures than the Zn/HA-DL coating. However, some potential micro-pores also exited at the Zn/HA interfaces in the Zn/HA composite layers. After immersion, oxidizations of Zn particles could seal the potential micro-pores in the Zn/HA composite layers. However, the electrochemical impedance between Zn particles and the electrolyte was lower than that between HA particles and the electrolyte. Therefore, the Zn/HA-ML coating exhibited a lower corrosion current and a higher corrosion resistance. Hope you can understand all contents.

Reviewer 2 Report

1. please explain the novelty of this work. This seems very trivial.

2. Too much of self citations and citations from a single country. This is really unethical. 

3. Please format the paper as per mdpi materials. 

4. Figure 2. XRD patterns - please mention the crystallographic planes. 

5. Conclusions are too qualitative. This needs to be made quantitative. 

6. Please show the exact samples and experimental facility. 

Minor changes required. 

Author Response

Response to Reviewer#2

Thank you for your valuable comments. We revised the manuscript according to your specific comments as follows. All corrections were underlined, which you could easily find in the revised article. The following is responses and explanations to your comments.

Comments#1: please explain the novelty of this work. This seems very trivial.

Responses: Thank you for your kind indications. Although the cold-sprayed Zn coating and Zn/HA layered coating for corrosion resistance of Mg alloy is reported, the Zn/HA composite coating prepared by high-velocity suspension flue spray (HVSFS) technology is never studied. This work about HVSF-sprayed Zn/HA composite layer on the cold-sprayed Zn layer proposed a potential for combining with merits of the two different technologies. The Zn/HA-ML coating effectively combined advantages of the corrosion resistance of CS Zn underlayer and the bioactivity of HVSFS Zn/HA multi-layers, which proposed a low-temperature strategy of improving corrosion resistance and bioactivity for implant metals.

Comments#2: Too much of self citations and citations from a single country. This is really unethical.

Responses: Thank you for your kind indications. Some references have been optimized as possible as we can. Hope you can understand all contents.

Comments#3: Please format the paper as per mdpi materials.

Responses: Thank you for your kind indications. The format of this paper has been revised according to the indication of this journal. Hope you can understand all contents.

Comments#4: Figure 2. XRD patterns - please mention the crystallographic planes.

Responses: Thank you for your kind indications. The crystallographic planes of different phases were shown in Figure 2 as possible as we can. Hope you can understand all contents.

Comments#5: Conclusions are too qualitative. This needs to be made quantitative.

Responses: Thank you for your kind indications. Some key results were quantitatively presented in the conclusion as possible as we can, which were highlighted in the revised manuscript. Hope you can understand all contents.

Comments#6: Please show the exact samples and experimental facility.

Responses: Thank you for your kind indications. Some exact samples and experimental facility were presented in section 2 as possible as we can, which were highlighted in the revised manuscript. Hope you can understand all contents.

Reviewer 3 Report

Several points need to be further clarified by the authors:

(1) It's beneficial to begin the abstract with a concise statement of the research purpose or problem. In this case, it could be something like "The study aims to enhance the corrosion resistance and bioactivity of Mg alloy substrates through the development of a Zinc/Hydroxyapatite (Zn/HA) multi-layer coating."

(2) The abstract provides a good overview of the methodology used. Please elaborate on the techniques or instruments used for analyzing phase, microstructure, and bonding strength to give readers a better understanding of the experimental approach.

(3) The sentence "Functionally graded or multi-layer coatings are successfully prepared by different technologies." in the introduction also requires the following reference: https://doi.org/10.1016/j.surfin.2022.102495; https://doi.org/10.3390/ma11030396.

(4) It's mentioned that the results indicate improvements in corrosion resistance and bioactivity. I suggest the abstract briefly highlights any significant findings or numerical data supporting these claims. This will make the abstract more informative.

(5) I encourage the authors to discuss the significance of their findings briefly. Why is the development of this Zn/HA multi-layer coating significant? How could it potentially impact the field of materials science or practical applications?

(6) In Table 2, the reported values for the cathodic branch slope (beta c) should be negative. Authors can put the ‘-’ sign behind the beta c in the first row.

(7) The presentation of the impedance data in the Nyquist plot is incorrect. The Nyquist plot's Z' and Z" axes should have the same scales; adding some characteristic frequencies to the diagrams is necessary. 

(8) The manuscript has minor grammatical issues and awkward phrasing. I suggest the authors review and revise these sections for clarity and conciseness.

(9) I suggest the conclusion ends with a brief concluding sentence summarizing the study's key takeaway. This can leave a lasting impression on readers.

(10) The manuscript mentions "Zn/HA multi-layer coating" several times. I encourage the authors to use synonyms or rephrase to avoid repetition.

(11) Ensure the MDPI journal's citation style for any references cited in the abstract.

(12) Include line numbers. This will help reviewers give more pointed feedback and help them assess whether the revisions are sufficient more quickly

Minor editing of English language required!

Author Response

Response to Reviewer#3

Thank you for your valuable comments. We revised the manuscript according to your specific comments as follows. All corrections were underlined, which you could easily find in the revised article. The following is responses and explanations to your comments.

Comments#1: It's beneficial to begin the abstract with a concise statement of the research purpose or problem. In this case, it could be something like "The study aims to enhance the corrosion resistance and bioactivity of Mg alloy substrates through the development of a Zinc/Hydroxyapatite (Zn/HA) multi-layer coating."

Responses: Thank you for your kind indications. A concise statement of the research purpose has been presented in the abstract, which was highlighted in the revised manuscript. Hope you can understand all contents.

Comments#2: The abstract provides a good overview of the methodology used. Please elaborate on the techniques or instruments used for analyzing phase, microstructure, and bonding strength to give readers a better understanding of the experimental approach.

Responses: Thank you for your kind indications. The methodologies in this paper have been presented in the abstract as possible as we can, which was highlighted in the revised manuscript. Hope you can understand all contents.

Comments#3: The sentence "Functionally graded or multi-layer coatings are successfully prepared by different technologies." in the introduction also requires the following reference: https://doi.org/10.1016/j.surfin.2022.102495; https://doi.org/10.3390/ma11030396.

Responses: Thank you for your kind indications. The references for this sentence have been added, which were highlighted in the revised manuscript. Hope you can understand all contents.

Comments#4: It's mentioned that the results indicate improvements in corrosion resistance and bioactivity. I suggest the abstract briefly highlights any significant findings or numerical data supporting these claims. This will make the abstract more informative.

Responses: Thank you for your kind indications. Some key data were presented in both the sections of abstract and conclusion, which were highlighted in the revised manuscript. Hope you can understand all contents.

Comments#5: I encourage the authors to discuss the significance of their findings briefly. Why is the development of this Zn/HA multi-layer coating significant? How could it potentially impact the field of materials science or practical applications?

Responses: Thank you for your kind indications. The significance of our findings were briefly discussed in section 3.6 as possible as we can, which was highlighted in the revised manuscript. Hope you can understand all contents.

Comments#6:  In Table 2, the reported values for the cathodic branch slope (beta c) should be negative. Authors can put the ‘-’ sign behind the beta c in the first row.

Responses: Thank you for your kind indications. The “-” sign has been added in the Table 2, which was highlighted in the revised manuscript. Hope you can understand all contents.

Comments#7: The presentation of the impedance data in the Nyquist plot is incorrect. The Nyquist plot's Z' and Z" axes should have the same scales; adding some characteristic frequencies to the diagrams is necessary.

Responses: Thank you for your kind indications. The EIS plots have been modified and as possible as we can as shown in Figure 6. Hope you can understand all contents.

Comments#8: The manuscript has minor grammatical issues and awkward phrasing. I suggest the authors review and revise these sections for clarity and conciseness.

Responses: Thank you for your kind indications. The English presentation through the article has been modified as possible as we can. Hope you can understand all contents.

Comments#9: I suggest the conclusion ends with a brief concluding sentence summarizing the study's key takeaway. This can leave a lasting impression on readers.

Responses: Thank you for your kind indications. This Zn/HA multi-layer coating proposed a novel strategy of improving corrosion resistance and bioactivity of implant metals. A brief concluding has been presented as possible as we can, which was highlighted in the revised manuscript. Hope you can understand all contents.

Comments#10: The manuscript mentions "Zn/HA multi-layer coating" several times. I encourage the authors to use synonyms or rephrase to avoid repetition.

Responses: Thank you for your kind indications. Some contents and figures have been improved and highlighted in the revised manuscript. Hope you can understand all contents.

Comments#11: Ensure the MDPI journal's citation style for any references cited in the abstract.

Responses: Thank you for your kind indications. The citation style for references has been modified according to the journal style, which was highlighted in the revised manuscript. Hope you can understand all contents.

Comments#12: Include line numbers. This will help reviewers give more pointed feedback and help them assess whether the revisions are sufficient more quickly

Responses: Thank you for your kind indications. Some contents and figures have been improved and highlighted in the revised manuscript. Hope you can understand all contents.

Reviewer 4 Report

Revised article deals with Zinc/Hydroxyapatite coating for corrosion protection of Mg alloy.  Deposited material is microstructurally characterized, while its corrosion behavior and biocompatibility were also studied. From the different variants assayed, the multi-layer coating demonstrated the best properties, directed towards e.g., bone implants. The description of the research done, the planning and the discussion of the results are correct. The extraction of conclusions is satisfactory. The novelty of the work, although it is not top level, has certain aspects of interest, as it is the potential application in prosthetics and implants.

Observations raised during evaluation are:

1.      Article needs to be completely formatted to MDPI style, i.e. references, and graphs/tables inserted in between the text.

2.      Pag 4. Define SBF acronym on its first appearance.

3.      Fig 4. Specify on Figure caption the source of the error bars.

4.      Pag 6. Section 3.4. Enlarge description of Tafel extrapolation method, including model equations used.

5.      Pag 6, Last sentence needs to be properly capitalized.

6.      Pag 7, first sentence on the page. Explain what phenomena is causing the increase in polarization resistance after immersion in test solution.

7.      Criteria used for assessing the correctness of EIS fittings.

8.      Equivalent circuit used for the EIS fittings. The physical significance of Rs is missing.

9.      Nyquist Impedance plots, Figs 6 (a), (b) and (c) need to be prepared with equal magnitude on both axis X and Y, e.g. 0-250 ohm·cm2 on axis X and axis Y (-Z''). This is the way to visually diagnose the performance of the system, to see if semicircles are flattened, quasi-symmetrical or stretched, and it corresponds to IUPAC recommendations (see Green book on Physical Chemistry).

10.   Tables 2 ad 3. State on Table caption the origin of the uncertainties appearing in the tables.

English needs checking for typos and minor details, but as prepared, the article is perfectly understandable.

Author Response

Response to Reviewer#4

Thank you for your valuable comments. We revised the manuscript according to your specific comments as follows. All corrections were underlined, which you could easily find in the revised article. The following is responses and explanations to your comments.

Comments#1: Article needs to be completely formatted to MDPI style, i.e. references, and graphs/tables inserted in between the text.

Responses: Thank you for your kind indications. The article has been formatted to MDPI style as possible as we can. Hope you can understand all contents.

Comments#2: Pag 4. Define SBF acronym on its first appearance.

Responses: Thank you for your kind indications. The SBF acronym has been modified in the revised manuscript. Hope you can understand all contents.

Comments#3: Fig 4. Specify on Figure caption the source of the error bars.

Responses: Thank you for your kind indications. The source of the error bars was the standard deviation of bonding strength after more than three testing for each condition. Hope you can understand all contents.

Comments#4: Pag 6. Section 3.4. Enlarge description of Tafel extrapolation method, including model equations used.

Responses: Thank you for your kind indications. The Tafel extrapolation method has been described as possible as we can, which was highlighted in the revised manuscript. Hope you can understand all contents.

Comments#5: Pag 6, Last sentence needs to be properly capitalized.

Responses: Thank you for your kind indications. The presentations have been modified as possible as we can. Hope you can understand all contents.

Comments#6:  Pag 7, first sentence on the page. Explain what phenomena is causing the increase in polarization resistance after immersion in test solution.

Responses: Thank you for your kind indications. The presentations have been modified as possible as we can. Hope you can understand all contents.

Comments#7: Criteria used for assessing the correctness of EIS fittings.

Responses: Thank you for your kind indications. EIS fittings were applied by Z-View2 software. The fitting accuracy is compared with the Chi-Square (chi-square test) data provided by ZView2 software. Chi-Square reflects the deviation between the measured value and the theoretical value of the statistical sample, and the degree of deviation determines the chi-square value. The larger the chi-square is, the more inconsistent the theory and the actual measurement are. When the chi-square value is 0, it shows that the theory and the actual measurement are completely consistent. Hope you can understand all contents.

Comments#8:  Equivalent circuit used for the EIS fittings. The physical significance of Rs is missing.

Responses: Thank you for your kind indications. Rs represent the ohmic resistance of the substrate and conductive lines. Hope you can understand all contents.

Comments#9: Nyquist Impedance plots, Figs 6 (a), (b) and (c) need to be prepared with equal magnitude on both axis X and Y, e.g. 0-250 ohm·cm2 on axis X and axis Y (-Z''). This is the way to visually diagnose the performance of the system, to see if semicircles are flattened, quasi-symmetrical or stretched, and it corresponds to IUPAC recommendations (see Green book on Physical Chemistry).

Responses: Thank you for your kind indications. The EIS plots have been modified as possible as we can as shown in Figure 6. Hope you can understand all contents.

Comments#10: Tables 2 ad 3. State on Table caption the origin of the uncertainties appearing in the tables.

Responses: Thank you for your kind indications. The standard deviation originated from more than three tests for each condition for Tables 2 and 3. Hope you can understand all contents.

Comments#11: Comments on the Quality of English Language

English needs checking for typos and minor details, but as prepared, the article is perfectly understandable.

Responses: Thank you for your kind indications. Some contents and figures have been improved and highlighted in the revised manuscript. Hope you can understand all contents.

Round 2

Reviewer 2 Report

Accept

Good

Reviewer 3 Report

I have reviewed the re-submission, and the authors have carefully amended their manuscript following the additional reviewers' suggestions.